# Antarctic Bottom Water Jets Flowing from the Vema Channel

**Eugene G. Morozov * , Oleg A. Zuev , Dmitry I. Frey and Viktor A. Krechik**

Shirshov Institute of Oceanology, Nakhimovsky Prospect 36, 117997 Moscow, Russia
* Correspondence: egmorozov@mail.ru

**Abstract:** Properties of the abyssal current of Antarctic Bottom Water (AABW) from the Vema Channel are studied based on temperature, salinity, and velocity profiler (CTD/LADCP) data. Previous studies over a period of almost 30 years revealed that very intense current of AABW exists in the Vema Channel. Later, it was found that this current consists of two branches. One branch spreads over the bottom of the channel; the other branch is elevated over the western wall of the channel. The deepest branch decays after it passes approximately 100 km while the upper one continues further to the North Atlantic and is the source of abyssal waters in the Canary and Cabo Verde basins of the North Atlantic. Data analysis suggested that the upper jet splits into two. One of these descends down a canyon at 24°30′ S, while the other (the third one) remains on the continental slope, and indications of its existence are also found at 24°00′ S. This research analyzes the existence and pathway of this third branch that can be traced up to latitude 24° S. Velocity measurements in 2022 allowed us to confirm the existence of this third branch.

**Keywords:** Antarctic Bottom Water; Vema Channel; CTD/LADCP measurements; three jets of bottom current





## 1. Introduction

Antarctic Bottom Waters originate from the Weddell Sea and spread to the north in the abyssal depths of the Atlantic Ocean. The Vema Channel is a pathway for the bottom waters through the Rio Grande Rise and Santos Plateau before they spread to the Brazil Basin and further to the north. Weddell Sea Deep Water (the densest abyssal water) flows only through the deep Vema Channel, which is approximately 4600–4800-m deep. The channel is a narrow passage situated between two terraces that are located on both sides. The narrowest channel width is about 15 km, and the channel length is about 700 km (Figure 1) [1].

Oceanographic studies of the Vema Channel have been carried out since the 1970s [2]. The largest number of CTD measurements was made over the section along 31°12′ S and at a point on this section at 39°18.3′ W. Since 1972, 29 visits have been made to the region; our group joined these studies in 2002 [3]. After 2005, CTD casts were supplemented with LADCP velocity profilers. The current of bottom water from the Vema Channel and circulation in the southern part of the Brazil Basin were studied for the first time in [4]. These studies revealed a strong current of AABW and displacement of the coldest core to the eastern wall of the channel [5]. A warming trend of Antarctic Bottom Water has been found in the Vema Channel [6–8]. The long-term temperature trend revealed on the basis of these data is shown in Figure 2. It can be observed that the temperature increase continues: our measurements taken in 2020 and 2022 reveal a continuing gradual increase in the bottom potential temperature at this point. The temperature increase is caused by the warming of Weddell Sea Deep Water in the Weddell Sea. The signal from the Weddell Sea to the Vema Channel has been propagating for more than 35 years [9]. Hence, a signal from the Weddell Sea, caused by warming approximately 40 years ago, has now been recorded. There are insufficient data to explain the abrupt change in temperature in the early 1990s as this process is related to the warming in the Weddell Sea in the 1950s–1980s.

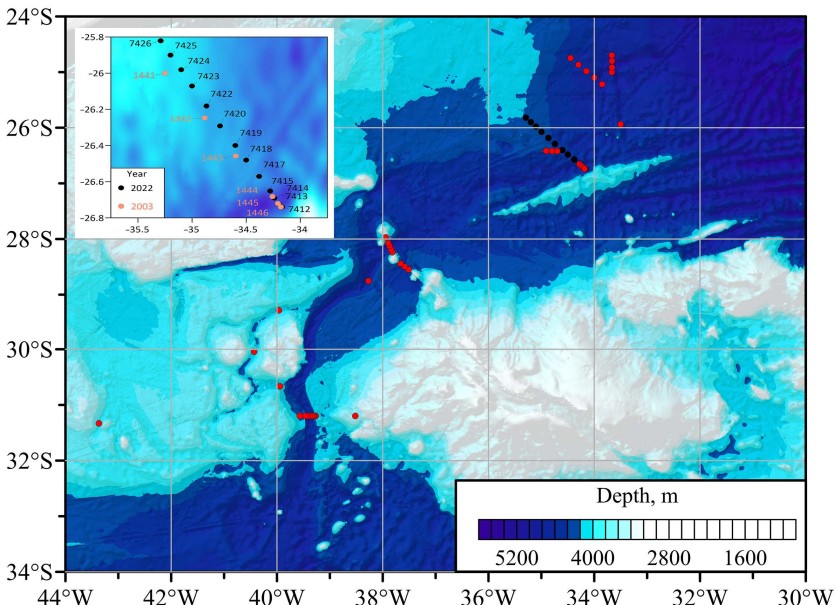

**Figure 1.** Bottom topography in the region of the Vema Channel (based on the GEBCO2019 data). Our stations in 2020 are indicated by red dots and those in 2022 are indicated by black dots. The inset shows stations in 2022 on a larger scale and stations in 2003 (dark yellow dots on the inset, not shown on the main map) approximately along the same line crossing the outflowing currents from the Vema Channel.

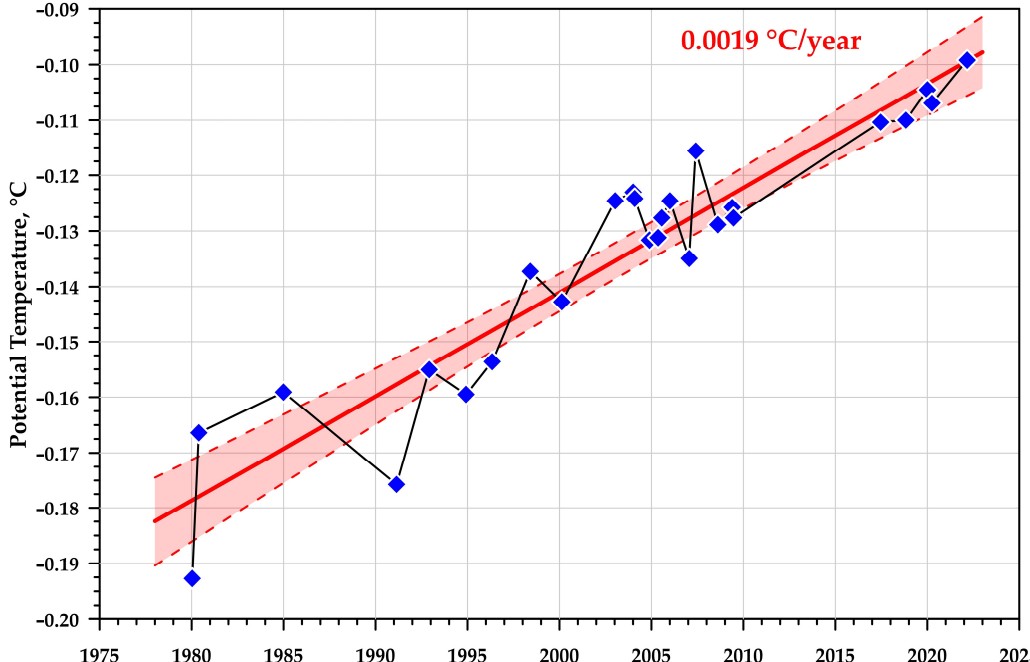

**Figure 2.** Long-term trend of potential temperature in the bottom layer of the Vema Channel. Blue dots indicate measurements during the visits for distinguishing the warming of AABW in the coldest part of the jet at 31°12′ S, 39°18.3 W. The fit curve indicates that the warming rate is 0.0019 °C per year. Red color between two dashed lines shows the 95% confidence interval.

The existence of two branches of Antarctic Bottom Water current in the Vema Channel was later reported in [10–12]. The AABW outflow from the Vema Channel was also studied based on a regional version of the ocean circulation model [13,14]. However, studies of the fine structure of the outflow require direct observations, which are extremely rare in this region. The pathways of bottom water propagation into the Brazil Basin north of the

Vema Channel have been studied based on the measurements of temperature, salinity, and velocity profilers (CTD/LADCP). It was suggested in [12] based on historical potential temperature measurements at the bottom that there is a third jet of the bottom water outflow. The objective of this work is to confirm the existence of this shallower jet (4200–4600 m) based on velocity measurements and map its pathway to the north up to 24° S.

## 2. Data

A long-term series of observations have been performed since the 1970s [3]. These measurements allow us to study long-term trends in the properties of the abyssal flow. Two regions of the channel were studied during the WOCE experiment in the 1990s. It is the standard section along 31°12′ S from 39°18′ W to 39°27′ W and the region in the northern part of the channel [15]. In addition, WOCE section A17 occupied by French scientists in 1994 and our stations in 2003 can be related to historical stations in the region. Complete information about the oceanographic stations with tables of coordinates and time is reported in [3]. Since this research concerns the northern part of the region, we emphasize that we performed investigations in the northern region of the Vema Channel in 2003, 2009, 2010, 2012, 2013, and, recently, in 2018, 2019, 2020, and 2022.

In cruise 87 of the R/V *Akademik Mstislav Keldysh* (AMK87) in 2022, we occupied one station over the section along 31°12′ S, which has been repeatedly occupied in the coldest jet at (31°12′ S; 39°18.3′ W). We also occupied a section of 13 stations north of the Vema Channel, which was planned to intersect all outflow jets and determine the position of individual stream jets of Antarctic Bottom Water outflow from the Vema Channel at latitudes 25.8–26.8° S. This section also made it possible to determine the flow velocities in the jets (Figures 1 and 3). The coordinates of stations in the region are given in Table 1.

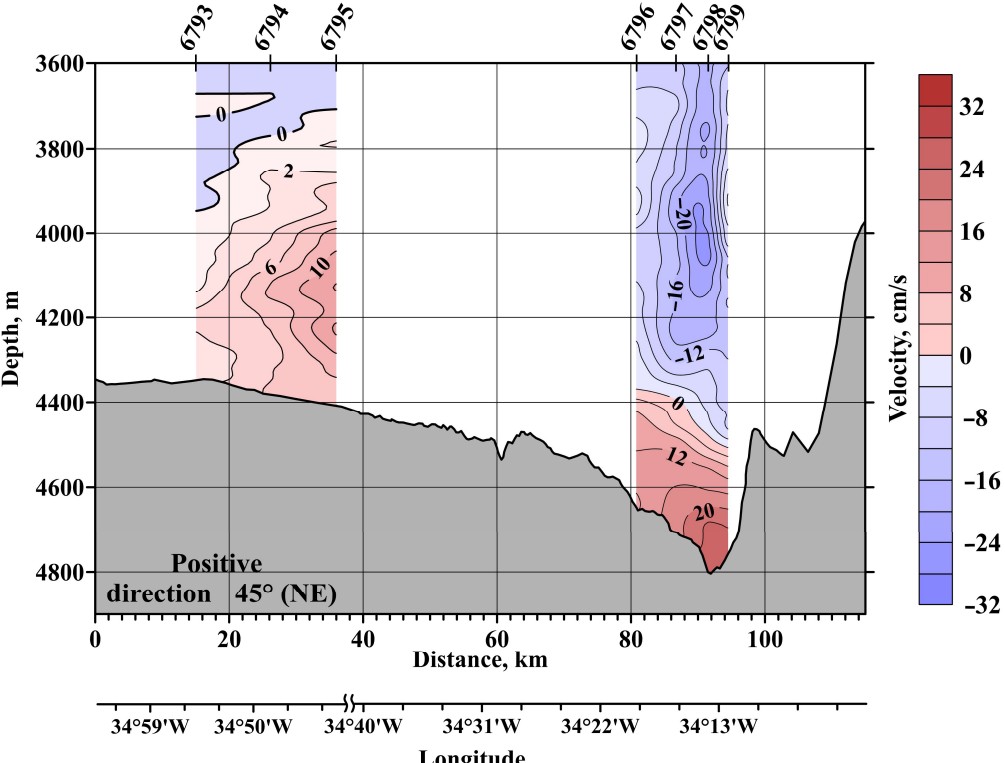

**Figure 3.** Distributions of velocity normal to the sections at 26°25′ S, 34°50′ W (western) and 26°40′ S, 34°15′ W (eastern) (see red dots in Figure 4) in April 2020. Numbers of stations are shown along the top axis [12].

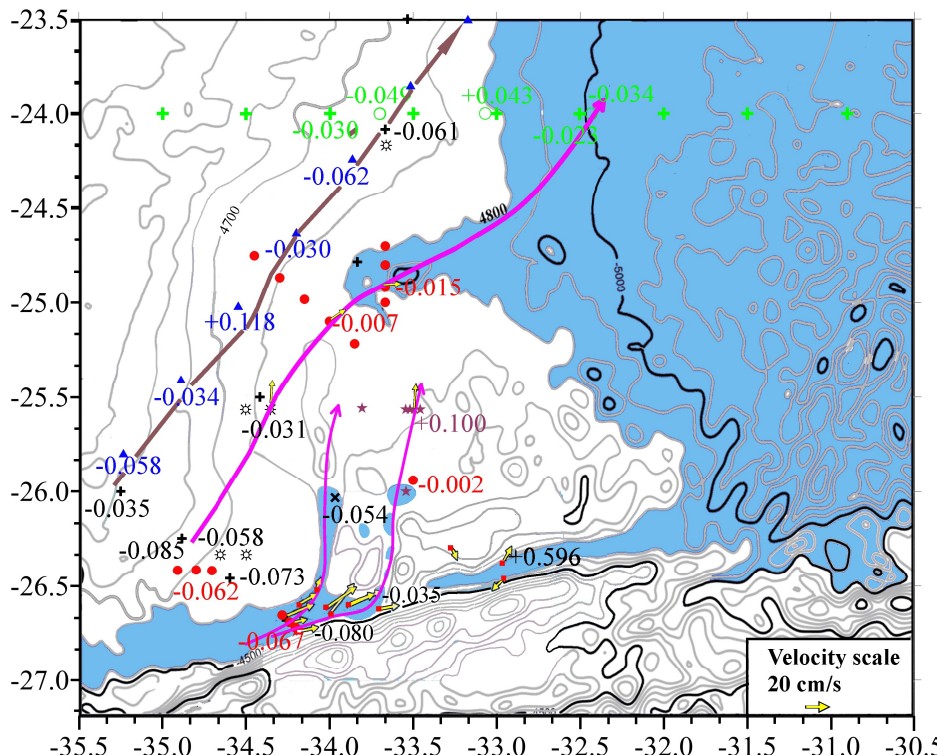

**Figure 4.** Bottom topography and stations in the study site. Bathymetry is based on GEBCO 2019, supplemented with our echo sounder measurements from 2010 to 2020. Locations of CTD/LADCP stations are indicated with different symbols. The bottom potential temperatures are indicated near the stations. Blue triangles show stations of the WOCE A17 transect in 1994. Black crosses (+) show stations in 2003. Sign (x) shows the stations in 2009. Green color shows stations and data of section A09 in 2009 and 2018 (24° S). Small red squares show stations in 2012 and 2013. Purple stars show stations in 2018. Circles with beams ( ✵ ) show stations in 2019. Red circles show stations in 2020. Yellow arrows show directions of currents and their speed (vectors). Deep water areas are shown in blue, especially the canyon at 24°40′ S. Two thin magenta lines show the previously analyzed AABW flows from the deep bottom of the channel. The thick magenta line shows the flow of the branch that initially was found above the western wall. The brown line shows the possible pathway of the shallowest jet (~4600–4700 m), which is plotted using only the CTD data from 1994 and 2003 [updated from [12]]. The existence of this flow was confirmed in 2022.

The Sea-Bird 19plus (Sea-Bird electronics, Bellevue WA, USA), Idronaut 320 plus (Idronaut S.R.L., Brugherio MS, Italy), and RDI Monitor 300 kHz (LADCP) (Teledyne RDI, San Diego, CA, USA) instruments were used for the measurements. The cold current jet was tracked based on the coldest bottom potential temperatures and velocity measurements using the LADCP profiler (Teledyne RDI, San Diego, CA, USA)at the stations where these data were available. It was also assumed that the current of bottom water should flow along the isobaths or go deeper.

The bottom topography used in this paper is based on satellite altimetry (GEBCO 2019) to plot our topographic maps. Some data of this digital bathymetry were corrected based on the measurements along the routes of our ships using the data of our echo sounders(Kongsberg Maritime, Kongsberg Viken, Norway).

**Table 1.** CTD/LADCP stations at the exit from the Vema Channel in 1994–2022.

| Stations | Date | Coordinates |
|---|---|---|
| | | R/V *Maurice Ewing* (only CTD) |
| 81 | 02.02.1994 | 27°21.1′ S, 36°36.7′ W |
| 82 | 02.02.1994 | 26°58.6′ S, 36°16.1′ W |
| 83 | 02.02.1994 | 26°34.6′ S, 35°55.7′ W |
| 84 | 02.02.1994 | 26°11.2′ S, 35°34.2′ W |
| 85 | 03.02.1994 | 25°47.8′ S, 35°14.0′ W |
| 86 | 03.02.1994 | 25°24.4′ S, 34°53.4′ W |
| 87 | 03.02.1994 | 25°01.1′ S, 34°32.7′ W |
| 88 | 03.02.1994 | 24°37.6′ S, 34°11.9′ W |
| 89 | 04.02.1994 | 24°14.2′ S, 33°51.8′ W |
| 90 | 04.02.1994 | 23°50.8′ S, 33°30.9′ W |
| 91 | 04.02.1994 | 23°27.5′ S, 33°10.3′ W |
| 92 | 04.02.1994 | 23°04.0′ S, 32°49.4′ W |
| | | R/V *Akademik Sergey Vavilov* (only CTD) |
| 1441 | 02.11.2003 | 26°00.0′ S, 35°15.0′ W |
| 1442 | 02.11.2003 | 26°14.9′ S, 34°53.0′ W |
| 1443 | 02.11.2003 | 26°27.5′ S, 34°35.8′ W |
| 1444 | 03.11.2003 | 26°40.9′ S, 34°15.4′ W |
| 1445 | 03.11.2003 | 26°43.2′ S, 34°12.1′ W |
| 1446 | 03.11.2003 | 26°44.4′ S, 34°10.8′ W |
| 1454 | 09.11.2003 | 27°05.5′ S, 35°54.7′ W |
| | | R/V *Akademik Ioffe* |
| 2079 | 18.04.2009 | 26°42.9′ S, 34°12.2′ W |
| 2080 | 18.04.2009 | 26°01.9′ S, 33°58.0′ W |
| 2437 | 06.11.2010 | 26°39.9′ S, 34°16.8′ W |
| 2438 | 06.11.2010 | 26°42.2′ S, 34°13.8′ W |
| 2439 | 06.11.2010 | 26°43.4′ S, 34°12.1′ W |
| | | R/V *Akademik Sergey Vavilov* |
| 2494 | 05.11.2012 | 26°36.8′ S, 33°59.3′ W |
| 2495 | 05.11.2012 | 26°35.9′ S, 34°10.2′ W |
| 2496 | 06.11.2012 | 26°31.1′ S, 34°03.3′ W |
| 2497 | 06.11.2012 | 26°35.8′ S, 33°51.7′ W |
| 2521 | 18.10.2013 | 26°20.5′ S, 32°00.1′ W |
| 2522 | 18.10.2013 | 26°27.7′ S, 32°53.1′ W |
| 2523 | 18.10.2013 | 26°23.0′ S, 32°53.1′ W |
| 2525 | 19.10.2013 | 26°18.6′ S, 33°11.5′ W |
| 2526 | 20.10.2013 | 26°35.7′ S, 33°51.6′ W |
| 2527 | 20.10.2013 | 26°37.3′ S, 33°39.6′ W |
| 2710 | 22.10.2018 | 25°34.1′ S, 33°29.5′ W |
| 2711 | 22.10.2018 | 25°34.0′ S, 33°31.1′ W |
| 2712 | 22.10.2018 | 25°34.0′ S, 33°27.4′ W |

**Table 1.** *Cont.*

| Stations | Date | Coordinates |
|---|---|---|
| 2713 | 22.10.2018 | 25°34.0′ S, 33°32.6′ W |
| 2714 | 23.10.2018 | 25°33.6′ S, 33°48.3′ W |
| 2716 | 23.10.2018 | 26°00.0′ S, 33°32.6′ W |
| | | R/V *Akademik Mstislav Keldysh* |
| 6563 | 30.12.2019 | 24°10.1′ S, 33°37.9′ W |
| 6564 | 30.12.2019 | 25°34.0′ S, 34°21.2′ W |
| 6565 | 31.12.2019 | 25°33.9′ S, 34°30.0′ W |
| 6566 | 31.12.2019 | 26°20.0′ S, 34°29.9′ W |
| 6567 | 31.12.2019 | 26°20.0′ S, 34°40.0′ W |
| 6793 | 03.04.2020 | 26°25.0′ S, 34°54.6′ W |
| 6794 | 03.04.2020 | 26°25.0′ S, 34°48.0′ W |
| 6795 | 03.04.2020 | 26°25.0′ S, 34°41.9′ W |
| 6796 | 06.04.2020 | 26°39.2′ S, 34°16.8′ W |
| 6797 | 06.04.2020 | 26°41.4′ S, 34°14.2′ W |
| 6798 | 06.04.2020 | 26°43.1′ S, 34°12.1′ W |
| 6799 | 06.04.2020 | 26°44.4′ S, 34°10.8′ W |
| 6800 | 07.04.2020 | 25°56.4′ S, 33°30.0′ W |
| 6801 | 07.04.2020 | 25°13.0′ S, 33°51.0′ W |
| 6802 | 07.04.2020 | 24°00.0′ S, 33°40.0′ W |
| 6803 | 07.04.2020 | 24°55.1′ S, 33°40.0′ W |
| 6804 | 07.04.2020 | 24°48.0′ S, 33°40.0′ W |
| 6805 | 08.04.2020 | 24°42.0′ S, 33°40.0′ W |
| 6806 | 08.04.2020 | 25°06.0′ S, 34°00.0′ W |
| 6807 | 08.04.2020 | 24°59.0′ S, 34°09.0′ W |
| 6808 | 08.04.2020 | 24°52.0′ S, 34°17.9′ W |
| 6809 | 08.04.2020 | 24°45.0′ S, 34°27.0′ W |
| 7411 | 02.03.2022 | 26° 54.1 S, 34° 18.8′ W |
| 7412 | 03.03.2022 | 26° 44.3 S, 34° 10.1′ W |
| 7413 | 03.03.2022 | 26° 43.2 S, 34° 12.0′ W |
| 7414 | 03.03.2022 | 26° 41.4 S, 34° 14.3′ W |
| 7415 | 03.03.2022 | 26° 39.0 S, 34° 16.7′ W |
| 7417 | 04.03.2022 | 26° 34.1 S, 34° 22.7′ W |
| 7418 | 04.03.2022 | 26° 28.7 S, 34° 30.0′ W |
| 7419 | 04.03.2022 | 26° 24.0 S, 34° 36.0′ W |
| 7420 | 04.03.2022 | 26° 17.4 S, 34° 44.5′ W |
| 7422 | 05.03.2022 | 26° 10.8 S, 34° 52.0′ W |
| 7423 | 05.03.2022 | 26° 04.2 S, 35° 00.0′ W |
| 7424 | 05.03.2022 | 25° 58.8 S, 35° 06.0′ W |
| 7425 | 05.03.2022 | 25° 54.0 S, 35° 12.0′ W |
| 7426 | 05.03.2022 | 25° 49.2 S, 35° 17.0′ W |

Our analysis is based on our CTD/LADCP measurements in 2003, 2004, 2009, 2010, 2012, 2013, 2018, 2019, 2020, and 2022. The casts reached depths approximately 5 m above the bottom. We also used the data from the WOCE A17 section (WHP, 2002) and the A09 section in 2009 and 2018 along 24° S (http://cchdo.ucsd.edu (accessed on 27 October 2022)). The data from several stations from the WOD18 database were also used. Velocity measurements were corrected by removing the currents of the barotropic tide using the TPXO 9 model [16].

## 3. Results

The CTD/LADCP measurements from the Vema Channel over sections in 2020 and earlier show the presence of the two cold AABW flows of Weddell Sea Deep Water (with potential temperatures below 0.2 °C). Both cold cores of the currents were observed in the sections at latitudes 26°30′–27°00′ S, measured in 2020 (Figure 3) [3,12]. Let us follow each jet and consider them separately. Each jet was identified based on the bottom topography (a channel deeper than the surroundings by 50–100 m), lower potential temperature, and velocities of the current.

The best-known AABW jet flows along the deep bed of the Vema Channel [5,17]. The deep core of the current is located between 4400 and 4700 m. Let us consider this deep outflowing current of AABW using the data gathered at our CTD/LADCP stations in 2003–2020. The stations were located in the region 26°30′–26°50′ S, 33°30′–34°20′ W (Figure 4). In 2003, the minimum measured potential temperature in this region of the northern part of the Vema Channel (Vema extension at 26°43.2′ S, 34°12.1′ W) was $\theta$ = −0.094 °C. In 2020, the potential temperature at the same point increased to $\theta$ = −0.067 °C. We found previously that the continuation of the coldest AABW flow splits into two jets [10,11]. These near-bottom jets rapidly warm and decay over a distance of about 100 km at latitude 25°30′ S (Figure 4).

Another AABW branch was detected over the northwestern slope of the Vema Channel. This flow is elevated by 600 m over the deepest branch of the AABW current [12]. The highest velocities in the core of this branch were recorded at depths of 4100–4200 m (150 m above the bottom). Such an isolated core of Weddell Sea Deep Water over the western terrace was repeatedly observed (in a section along 31°12′ S) from 1984 to 2020. The potential temperature of this branch has been increasing over time. The measurements of velocities within 30–36 cm/s using LADCP over the repeated section confirm the stable existence of this branch [3,18]. The continuation of this jet was found at the outflow of the Vema Channel in 2020 at 26°25′ S, 34°42′ W above a depth of 4420 m (Figure 3, western section). Then, this jet continues to the north, approaches the upper part of a zonal canyon at 24°30′ S (Figure 4), and flows down the canyon to depths deeper than 4800 m [12].

In 2009, scientists from Great Britain occupied a WOCE CTD section (A09) along 24° S. They repeated experiments on this section in 2018 (https://cchdo.ucsd.edu, accessed on 27 October 2022). In 2009, two cores of cold water were detected in this section: the bottom potential temperatures at longitude 31°50′ W (at depths of approximately 4650–4750 m) ranged from −0.04 °C to −0.05 °C, while at 33°50′ W (in the depth range of 5000–5200 m), potential temperatures ranged from −0.03 °C to −0.04 °C. These cores were slightly displaced in 2018. Potential temperatures in the cold cores in 2018 were close to −0.03 °C. Sections of potential temperature along 24° S in 2009 and 2018 are shown in Figure 5. No measurements of currents over this section are available.

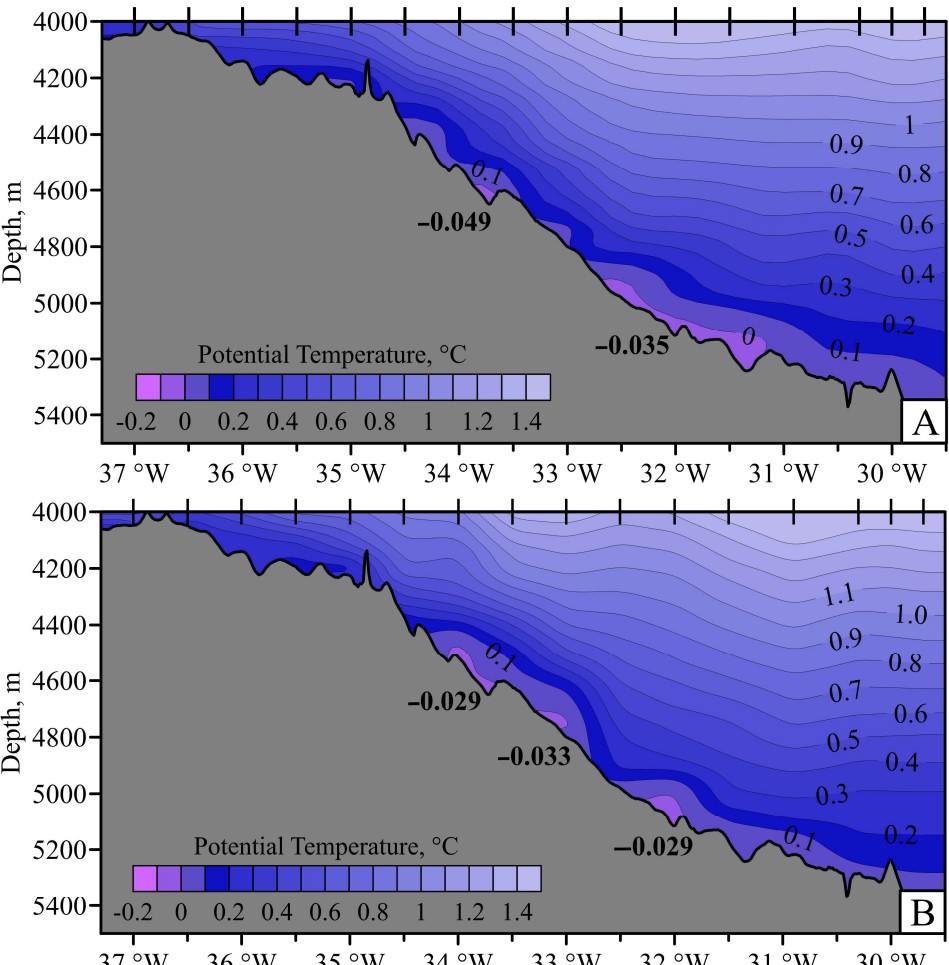

**Figure 5.** Sections of potential temperatures deeper than 4000 m in 2009 (**A**) and 2018 (**B**) (WOCE line A09 along 24° S). The coldest bottom potential temperatures are indicated over the background of the bottom.

In Figure 5, two cores of cold water are observed at depths of 4600–4800 m and 5100 m. Previously, it was found that two jets of bottom current became warmer and decayed at 25°30′ S at depths of ~4700 m [12]. Thus, they could not transport very cold water to latitude 24° S. The jet that descended the canyon could have been recorded at 24° S at a depth of 5100 m. However, a cold core at 33°–34° W at depths of 4700 m could appear only if this water has been transported by another jet. Finding this jet is the goal of this research. Below, the existence of this jet is considered in more detail. The potential temperature of cold water detected at 24°00′ S, 33°42′ W in 2009 at depths of ~4600 m was −0.049 °C. By 2018, it had become warmer (−0.029 °C).

There is no other source of such cold water than the upper branch of the current from the Vema Channel. The section at 24° S that was occupied in 2009 and 2018 is located approximately 300 km north of the Vema Channel. The continental slope of South America here is not at all steep: the slope is estimated as 1/(100–120). A moving flow along the isobaths can keep its path above such a slope due to the balance of the gravity force directed down the slope and the Coriolis force that is directed to the left of the flow upslope. Small channels on the continental slope with a depth of about 50–100 m over a section at 24° S were found exactly coinciding with the cold potential temperatures above the bottom. Thus, these flows could have eroded them in the sediments. In our expeditions after 2018, we tried to find the pathway of the continuation of the upper jet north of the Vema Channel. The criterion was the existence of low bottom potential temperatures and a general northward direction of velocity, with possible deviations along the isobaths to the east.

The upper branch of the AABW current was traced on the basis of low potential temperatures and the northern-northeastern direction of velocity vectors at the bottom. These vectors are shown by the yellow arrows in Figure 4. In 2020, we found that this bottom branch descends into an underwater abyssal canyon (Figure 4), which is at approximately 24°25′–24°40′ S [12]. At longitude 33°10′ W, this canyon becomes wider. One can see from the historical bottom temperature distribution that this canyon diverts part of this cold AABW jet downslope. The branch that descends the canyon was detected in 2020 based on low potential temperatures and vectors of velocities at the bottom at five locations between 26°25′ S and 24°55′ S. This branch forms the cold core at a depth of 5100 m at 24° S.

At latitude 24° S, one can see another cold core at depths of 4600 at longitudes 33°–34° W. This core could have been formed by a jet located shallower than the one that descended the canyon. Therefore, there is a third jet at depths of ~4600 m that has been detected from low potential temperatures over the section. This branch has been traced from low temperatures based on CTD casts in 1994 and 2003; however, no LADCP measurements were made in 1994 or 2003. In 1994–2003, the bottom potential temperatures were much lower than now. This jet is drawn in Figure 4 with a brown line connecting the points, with cold potential temperatures at the bottom. When plotting this line, we tried to connect measurement points with low temperatures. In addition, the pathway was drawn so that over steep slopes the flow was descending more rapidly than over flatter slopes. The potential temperature of the bottom flow has been strongly increasing since 1994. The very cold temperatures that were measured in 1994 and 2003 have not since been recorded near the bottom. Thus, we detected a third branch of AABW flow directed to the north, which is shown in Figure 5 as a cold core at 24° S at depths of 4600 m. The potential temperature of this upper branch at a depth of 4600 m is below zero at 24° S. Based on the measurements in the databases north of 24° S, this branch spreads further to the north. The continuation of this jet north of 24° S is presumably the flow of AABW described in [19]. Thus, on the basis of our targeted measurements and historical data, we have constructed a scheme of pathways for the continuation of AABW flow (Figure 4).

To detect this jet at the outflow from the Vema Channel, we constructed a section in 2022 to cross all possible flows from the Vema Channel. Before measurements were taken in 2022, a CTD section (without LADCP) was constructed in 2003 of only five stations (1441–1446) (Figures 6 and 7) approximately along the same line (inset in Figure 1). Section 2003 does not allow us to resolve three jets of currents because of a small number of stations and lack of LADCP measurements. Because of the general warming of abyssal waters in the Atlantic, potential temperatures increased significantly [8].

The section of 13 stations occupied in 2022 revealed three branches of AABW current from the Vema Channel, which were presumably detected from previous measurements. These jets are observed both on the temperature section and on the sections of the current components along the meridian and normal to the section. Jets of currents on a gentle slope are observed at longitudes of 35.1° W and 34.7° W.

Three different components of currents across the section north of the Vema Channel are shown in Figure 8. The currents are generally directed to the east-northeast. High current velocities are observed at the bottom in the channel bed closer to the western wall: up to 21 cm/s with an eastern-northeastern direction at station 7415. At the neighboring station 7417, the current accelerates to 28 cm/s and acquires a pronounced northern direction near the bottom. A fairly wide flow with velocities exceeding 10 cm/s and a predominant direction to the east-northeast was noted on a gentle slope. The maximum velocities (20 cm/s) near the bottom were simultaneously recorded at station 7424 (35°06′ W) in the same place where the third jet with the minimum potential temperature was observed. Even higher current velocities, up to 29 cm/s, were noted in the depth range of 3800–3900 m at stations 7420, 7424, and 7426. Our previous numerous LADCP casts that returned profiles of bottom currents in the Vema Channel suggest that the maximum velocities were found not at the very bottom but 100–150 m above it, while the water with the coldest potential temperatures was detected exactly at the bottom.

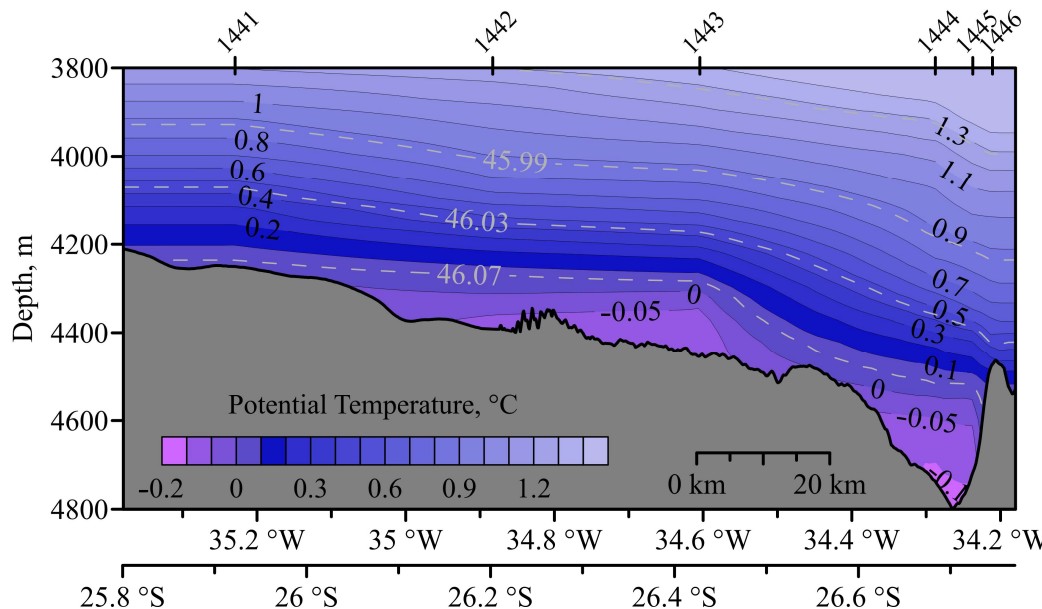

**Figure 6.** Potential temperature section north of the Vema Channel in 2003. Contour lines of density (~46 units) are shown.

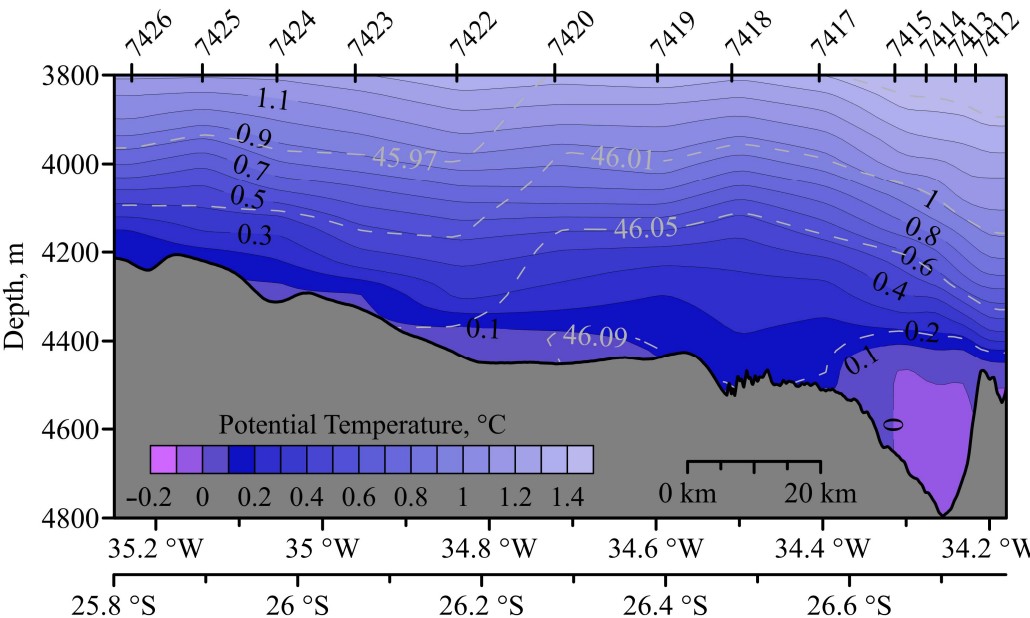

**Figure 7.** Potential temperature section of AABW outflow from the Vema Channel in 2022. The location of the section was slightly different from its position in 2003 (Figure 1). Contour lines of density (~46 units) are shown.

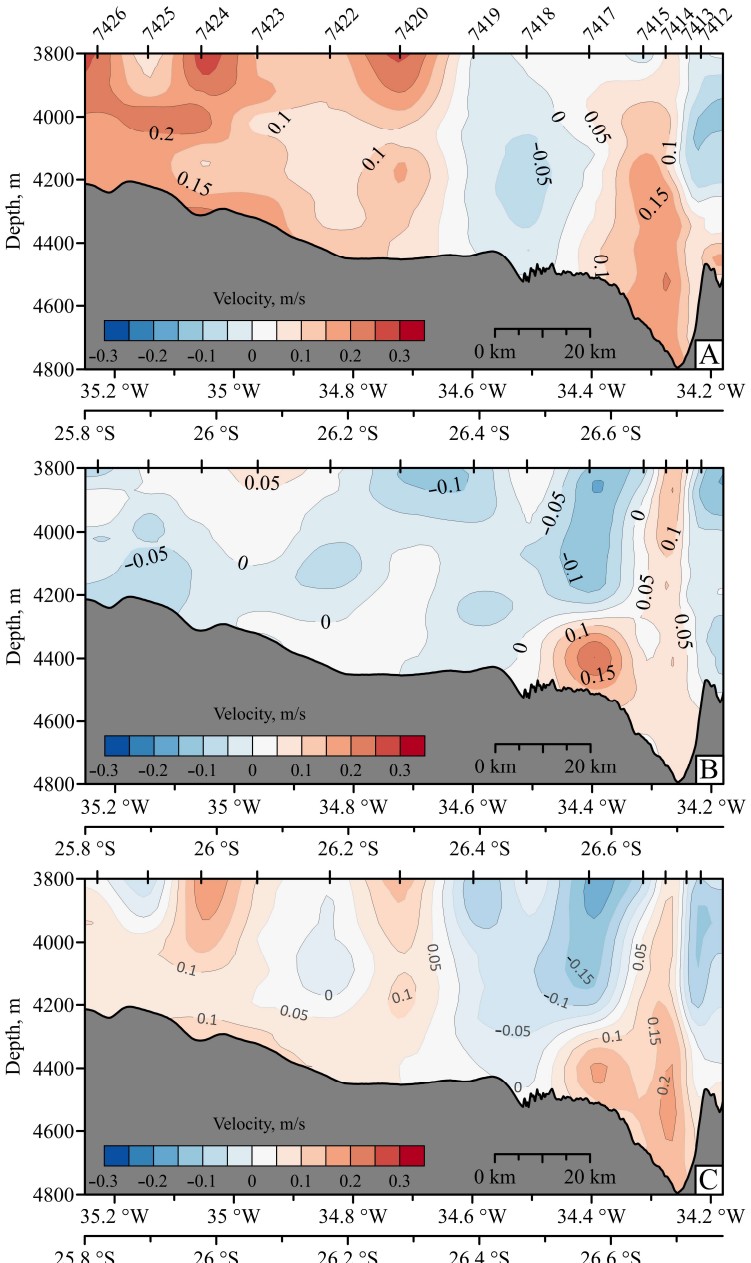

**Figure 8.** Sections of the velocity components at the outflow of the AABW from the Vema Channel in 2022. Top panel (**A**): positive direction to the east; middle panel (**B**): positive direction to the north; bottom panel (**C**): positive direction to the northeast, normal to the section.

Thus, our analysis reveals the existence of three jets of bottom flow that continue the spreading of AABW in the Vema Channel. Previously, the existence of a third jet before measurements were taken in 2022 could only be judged from CTD data. Presumably, the upper jet, which initially flows above the western wall of the Vema Channel, splits into two branches of continuation, and both of them spread further north than the lower jet from the deep bottom of the Vema Channel. Therefore, the upper jet is the current that transports AABW further north and is the source of abyssal bottom water in the deep basins in the North Atlantic. The shallower part of this upper flow (the third jet in our context) transports cold water to 24° S and is observed as the cold core at depths of 4600 m. The deeper part of the upper jet descends the canyon and fills the deepest abyssal regions of the Atlantic.

**Author Contributions:** E.G.M.: conceptualization and original draft preparation; D.I.F., V.A.K. and O.A.Z.: field data, original draft preparation, writing, and figures. All authors have read and agreed to the published version of the manuscript.

**Funding:** The work was supported by the Russian Science Foundation grant 21-77-20004.

**Institutional Review Board Statement:** Not applicable.

**Informed Consent Statement:** No studies involving humans and animals were performed.

**Data Availability Statement:** Data are available upon request.

**Conflicts of Interest:** The authors declare no conflict of interest.

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
