# Peer review of "Antarctic Bottom Water Jets Flowing from the Vema Channel"

_water, doi:10.3390/w14213438_

Round 1
Reviewer 1 Report
The paper reports analysis of data in a region of the ocean where, despite its high relevance, in situ observations are extremely rare. The results contain original information and constitute an important contribution to understand the pathways of Antarctic Bottom Water flow through the Vema Channel and the modification it undergoes before continuing northward, into the Brazil Basin. I read the manuscript carefully and did not find any flaw that would impede its publication by Water. Therefore, I enthusiastically recommend its acceptance in the present form.
Author Response
Thank you for the high opinion about our manuscript
Reviewer 2 Report
I tried several times to read this manuscript, and repeatedly failed. The idea looks interesting enough, but the presentation need a lot of work. The problem is not English language issues per se, but rather a lack of organization. The intro reads like a data section, then when one gets to the data section, the reader is confused with a list of dates and lat-long positions. There are also many constructions such as "We observed", etc, where it is not clear whether this refers to we in the present (e.g. the 2022 cruise) or in the distant past. The figures help, but even there, the reader is lost as to which station dots are what year, which were re-occupied, etc. Perhaps a table and more careful attention to the figures would help. For example, Fig 3 could be an inset on Fig 1. Basically,
i) for the intro, just describe the current understanding of how AABW flows through Vema Channel and beyond, plus maybe the warming, but without too much tedious detail, leading up to the last few sentences where you define the main goals of this work.
ii) Data section needs better organization. List all the data used, and then single out the particular data sets that make up the added value of this paper.
iii) How the pathways (e.g., Fig 4) are obtained needs a cleaner explanation.
Where I gave up was p. 7. It is very difficult to follow the flow of ideas and how they relate to the figures shown.
The English could also use a read-through by a native speaker, if you can find one. Avoid overuse of the second person (we) and when referring to your past work, give a reference. I believe that there will be significant interest in this work, but not before it is more clearly presented.
Round 2
Reviewer 2 Report
I thought the manuscript reads much better this time around and now recommend publications. A few minor comments follow:
Abstract: line 8: ... are studied using temperature, salinity, and velocity profiler (CTD/LADCP) data.
l. 71: ...confirm the existence of this shallower jet...
l. 195: Something is wrong with this sentence. Do you mean "We traced the upper branch?
l. 204: ...deepest branch, which flows from ... Channel, splits in two.
l. 209: It is difficult to keep track of all the various branches here. As the paragraph ending on line 209 ends, I thought there were three, but then the next paragraph introduces another, and calls this the third branch. Please clean up the writing in these two paragraphs.
Author Response
We thank the reviewer and editor for their time and efforts to improve the paper.
We have accepted all corrections suggested by the reviewer and editor.